# Combining simple blood tests to identify primary care patients with unexpected weight loss for cancer investigation: Clinical risk score development, internal validation, and net benefit analysis

**Brian D. Nicholson**[1]*, **Paul Aveyard**[1], **Constantinos Koshiaris**[1], **Rafael Perera**[1], **Willie Hamilton**[2], **Jason Oke**[1], **F. D. Richard Hobbs**[1]

1 Nuffield Department of Primary Care Health Sciences, University of Oxford, Oxford, United Kingdom,
2 Medical School, University of Exeter, Exeter, United Kingdom

* brian.nicholson@phc.ox.ac.uk

## Abstract

### Background

Unexpected weight loss (UWL) is a presenting feature of cancer in primary care. Existing research proposes simple combinations of clinical features (risk factors, symptoms, signs, and blood test data) that, when present, warrant cancer investigation. More complex combinations may modify cancer risk to sufficiently rule-out the need for investigation. We aimed to identify which clinical features can be used together to stratify patients with UWL based on their risk of cancer.

### Methods and findings

We used data from 63,973 adults (age: mean 59 years, standard deviation 21 years; 42% male) to predict cancer in patients with UWL recorded in a large representative United Kingdom primary care electronic health record between January 1, 2000 and December 31, 2012. We derived 3 clinical prediction models using logistic regression and backwards stepwise covariate selection: Sm, symptoms-only model; STm, symptoms and tests model; Tm, tests-only model. Fifty imputations replaced missing data. Estimates of discrimination and calibration were derived using 10-fold internal cross-validation. Simple clinical risk scores are presented for models with the greatest clinical utility in decision curve analysis. The STm and Tm showed improved discrimination (area under the curve $\geq$ 0.91), calibration, and greater clinical utility than the Sm. The Tm was simplest including age-group, sex, albumin, alkaline phosphatase, liver enzymes, C-reactive protein, haemoglobin, platelets, and total white cell count. A Tm score of 5 balanced ruling-in (sensitivity 84.0%, positive likelihood ratio 5.36) and ruling-out (specificity 84.3%, negative likelihood ratio 0.19) further cancer investigation. A Tm score of 1 prioritised ruling-out (sensitivity 97.5%). At this threshold, 35 people presenting with UWL in primary care would be referred for investigation for each

**Data Availability Statement:** This study is based on CPRD data and is subject to a full licence agreement which does not permit data sharing outside of the research team.

**Funding:** BDN was supported by National Institute for Health Research (NIHR) Doctoral Research Fellowship number (DRF-2015-08-18). FDRH and RP acknowledges part-funding from the NIHR Oxford Medtech and In-Vitro Diagnostics Co-operative (MIC). FDRH, RP, and PA acknowledge part-funding from the NIHR Oxford and Thames Valley Applied Research Collaboration (ARC). FDRH, JO, RP, and PA acknowledge part-funding from the NIHR Oxford Biomedical Research Centre (BRC). PA is an NIHR senior investigator. RP acknowledges part-funding from the National Institute for Health Research (NIHR Programme Grant for Applied Research) and the Oxford Martin School. WH is co- Principal Investigator of the multi-institutional CanTest Research Collaborative funded by a Cancer Research UK Population Research Catalyst award (C8640/A23385). The views expressed are those of the authors and not necessarily those of the NHS, the NIHR, or the Department of Health. The funders had no role in study design, data collection and analysis, decision to publish, or preparation of the manuscript.

**Competing interests:** The authors have declared that no competing interests exist.

**Abbreviations:** AUC, area under the curve; CPRD, Clinical Practice Research Datalink; DCA, decision curve analysis; EHR, electronic health record; ISAC, Independent Scientific Advisory Committee; NLR, negative likelihood ratio; NPV, negative predictive value; ONS, Office for National Statistics; PLR, positive likelihood ratio; PMM, predictive mean matching; PPV, positive predictive value; Sm, symptoms-only model; SNB, standardised net benefit; STm, symptoms and tests model; Tm, tests-only model; UWL, unexpected weight loss.

person with cancer referred, and 1,730 people would be spared referral for each person with cancer not referred. Study limitations include using a retrospective routinely collected data-set, a reliance on coding to identify UWL, and missing data for some predictors.

## Conclusions

Our findings suggest that combinations of simple blood test abnormalities could be used to identify patients with UWL who warrant referral for investigation, while people with combinations of normal results could be exempted from referral.

## Author summary

### Why was this study done?

- The risk of an early and late stage cancer diagnosis is increased during the 3 to 6 months following the first record of unexpected weight loss (UWL) in primary care. UWL presents a diagnostic challenge as it is associated with a wide range of other benign and serious conditions.

- Diagnostic strategies that avoid the harms of unnecessary invasive and costly cancer investigation are required for patients with UWL. Our research has shown that the absence of individual or pairs of co-occurring clinical features does not reduce the likelihood of cancer enough to sufficiently rule-out patients from further cancer investigation. It has also identified that primary care clinicians commonly request multiple blood tests when patients present with UWL.

- We aimed to identify whether the presence or absence of risk factors, symptoms, signs, and blood test results could be used together to rule-out more accurately the need for cancer investigation in patients with UWL.

### What did the researchers do and find?

- We analysed the electronic health records of 63,693 adults with UWL recorded between January 1, 2000 and December 31, 2012 to derive 3 clinical scores including symptoms, symptoms and test results, and test results, to predict the risk of cancer within 6 months.

- The scores including test results were discriminative between patients with and without cancer, were well calibrated at the levels of risk that decisions to investigate are made in primary care, and showed superior clinical utility compared to the symptoms-only model.

### What do these findings mean?

- Simple scores including age-group, sex, and 7 simple primary care blood tests (albumin, alkaline phosphatase, C-reactive protein, haemoglobin, liver enzymes, platelets, and

total white cell count) could be used to select patients with UWL who do not warrant further cancer investigation in addition to those that do.

- Further research is required to validate these scores in external datasets from settings, populations, and subgroups of interest, to understand how to maximise uptake in primary care, and to assess whether the use of this approach to cancer investigation might impact cancer outcomes.

## Introduction

Unexpected weight loss (UWL) is a presenting feature of cancer for which there remains no consensus on the most appropriate investigation strategy in primary care [1]. Patients with UWL recorded by their primary care clinician are more likely to be diagnosed with the following cancers within 3 months: pancreatic, cancer of unknown primary, gastro-oesophageal, lymphoma, hepatobiliary, lung, bowel, and renal tract [2]. This association is greatest in males once aged 60 years or older and in females 80 years or older [2,3]. Current investigation guidelines focus on selecting patients for single-site cancer investigation based on simple combinations of clinical features (individual risk factors, signs, symptoms, and blood test abnormalities) [3–5].

As most patients presenting to primary care with UWL will not have cancer, diagnostic strategies that avoid the harms of unnecessary invasive and costly investigation are also required for patients at a low risk of cancer [1]. Our previous work has shown that the presence of individual co-occurring clinical features increases the likelihood of cancer sufficiently to rule-in cancer investigation [5]. However, the absence of individual co-occurring clinical features, including pairs of normal inflammatory markers, do not reduce the likelihood of cancer sufficiently enough to rule-out patients from further cancer investigations [5,6]. Primary care clinicians commonly request multiple blood tests when patients present with UWL [5,7]. There is little guidance on how clinicians should interpret these blood tests in combination or which are most relevant for use in clinical practice [1,6]. When baseline investigations are normal, a watchful waiting approach may be preferable to invasive testing [8].

Prediction models have been developed to identify the most helpful combinations of clinical features for use in clinical practice [9,10]. However, these studies were based on small cohorts from secondary care; they recommend conflicting approaches and include some investigations uncommon in primary care. Research using data from primary care is therefore needed to investigate whether the absence of risk factors and co-occurring clinical findings in the context of normal test results could reduce the risk of cancer to sufficiently rule-out patients with UWL from invasive cancer investigation.

We aimed to derive and internally validate prediction models using co-occurring risk factors, symptoms, signs, and blood test data to identify those clinical features that could be used together to stratify cancer risk in patients attending primary care with UWL.

## Methods

The protocol was approved by the Independent Scientific Advisory Committee (ISAC) of the MHRA (protocol number 16_164A2A) [11]. Ethics approval for observational research using the CPRD with approval from ISAC was granted by a National Research Ethics Service committee (Trent Multiresearch Ethics Committee, REC reference number 05/MRE04/87). We

followed the TRIPOD (S1 TRIPOD Checklist) reporting guidelines [12]. Stata (version 15) was used for all analyses.

## Cohort design and population

We selected a cohort of patients with UWL indicated by the presence of a code for UWL previously been shown to be linked to measured weight loss [13,14]. Patients were selected for the derivation cohort if UWL was first coded between January 1, 2000 and December 31, 2012 in the Clinical Practice Research Datalink (CPRD). The CPRD is an anonymised database of primary care records database covering a representative 6.9% of the United Kingdom population [15]. Patients were included if they were ≥18 years of age, registered with a CPRD general practice, eligible for linkage to NCRAS and Office for National Statistics (ONS) data, and at least 12 months of data before their first UWL code (the "index date"). These UWL Read codes equated to a mean weight loss of ≥5% within a 6-month period in our previous internal validation study of weight-related coding in CPRD [13]. UWL may be coded following a range of clinical scenarios, including UWL reported as the patient's presenting complaint, after targeted history taking, following weight measurement as part of the clinical examination, or as part of a routine health check or chronic disease review [5]. Patients were excluded if they had a prescription of weight-reducing medication (orlistat) or a code for bariatric surgery in the previous 6 months, or if they had been previously diagnosed with cancer.

## Outcome definition

The outcome was any cancer diagnosed within 6 months of the index date identified in the CPRD or NCRAS, using an existing library of codes [2]. Patients were followed up until the date of the first cancer diagnosis or for 6 months, whichever occurred first. Six months was chosen as previous research has shown that this is the period associated with an increased risk of cancer diagnosis following a presentation of UWL to primary care [2]. Cancers classified as nonmelanoma skin cancer, in situ, benign, ill-defined, or uncertain were excluded.

## Predictor variables

Sociodemographic features, recorded on or before the index date, were extracted for each patient (Table 1). Preexisting comorbidities were identified using a previously described approach [13]. Clinical features shown to be associated with cancer when recorded in the 3 months before to 1 month after the UWL date were identified within that time period [5]. Continuous results of blood tests commonly requested within this time period were also identified using entity codes in CPRD, and outliers and erroneous results were dropped [5] (Table 2).

## Multiple imputation

Multiple imputation was used to replace missing values for smoking status, alcohol intake, body mass index, and blood tests using the mi suite of commands in Stata [16,17]. Fifty imputed datasets were created. Multinomial logistic regression was used to impute categorical variables, and predictive mean matching (PMM) with 5 donors was used to impute continuous variables [18]. The imputation model included the outcome, all candidate variables to be included in the final predictions models, and auxiliary variables to increase the likelihood that the missing at random assumption was satisfied. These were a combination of variables found to predict missingness, personal characteristics, comorbidities, risk factors, other markers of inflammation, or full blood count components (S1 Text). For the primary analysis, continuous

**Table 1. Baseline characteristics of the derivation cohort and imputation dataset.**

| | Derivation cohort | | Imputation dataset |
|---|---|---|---|
| | n (%) | mean (SD) | %/mean (SD) |
| Sex | | | |
| Male | 26,758 (41.8) | | 41.8 |
| Female | 37,215 (58.2) | | 58.2 |
| Age-group (years) | | | |
| 18–39 | 14,290 (22.3) | | 22.3 |
| 40–49 | 8,016 (12.5) | | 12.5 |
| 50–59 | 8,511 (13.3) | | 13.3 |
| 60–69 | 9,017 (14.1) | | 14.1 |
| 70–79 | 11,565 (18.1) | | 18.1 |
| 80+ | 12,574 (19.7) | | 19.7 |
| Smoking status | | | |
| Current | 9,629 (15.1) | | 31.0* |
| Ex-smoker | 7,164 (11.1) | | 22.9* |
| Nonsmoker | 14,457 (22.6) | | 46.0* |
| Missing | 32,723 (51.2) | | - |
| Alcohol consumption | | | |
| Current | 18,435 (28.8) | | 66.9* |
| Nondrinker | 8,095 (12.7) | | 29.1* |
| Past drinker | 1,087 (1.7) | | 4.0* |
| Missing | 36,356 (56.8) | | - |
| Body mass index | | | |
| Underweight | 6,691 (10.9) | | 12.4* |
| Normal | 33,846 (52.9) | | 60.0* |
| Overweight | 10,790 (16.9) | | 18.8* |
| Obese | 5,141 (8.0) | | 8.8* |
| Missing | 7,235 (11.3) | | - |
| Comorbidities | | | |
| 0 | 8,870 (13.9) | | 13.9 |
| 1 | 12,762 (20.0) | | 20.0 |
| 2 | 12,641 (19.8) | | 19.8 |
| 3 | 10,378 (16.2) | | 16.2 |
| 4 | 7,638 (11.9) | | 11.9 |
| 5+ | 11,684 (18.3) | | 18.3 |
| Co-occurring clinical features | | | |
| Appetite loss | 1,592 (2.5) | | 2.5 |
| Abdominal mass | 233 (0.4) | | 0.4 |
| Abdominal pain | 3,762 (5.9) | | 5.9 |
| Back pain | 3,277 (5.1) | | 5.1 |
| Change in bowel habit | 641 (1.0) | | 1.0 |
| Chest pain (noncardiac) | 1,866 (2.9) | | 2.9 |
| Chest signs | 90 (0.1) | | 0.1 |
| Dyspepsia | 1,776 (2.8) | | 2.8 |
| Dysphagia | 717 (1.1) | | 1.1 |
| Haemoptysis | 181 (0.3) | | 0.3 |
| Iron deficiency anaemia | 780 (1.2) | | 1.2 |
| Jaundice | 126 (0.2) | | 0.2 |

*(Continued)*

**Table 1.** (Continued)

| | Derivation cohort | | Imputation dataset |
|---|---|---|---|
| | n (%) | mean (SD) | %/mean (SD) |
| Lymphadenopathy | 209 (0.3) | | 0.3 |
| Venous thromboembolism | 201 (0.3) | | 0.3 |
| Liver function tests | | | |
| Albumin (g/L) | 41,622 (65.1) | 41.1 (4.8) | 41.2 (4.8)* |
| Alkaline phosphatase (iu/L) | 41,278 (64.5) | 90.8 (57.5) | 89.9 (56.8)* |
| AST/ALT (iu/L) | 10,911 (17.1) | 25.9 (20.7) | 25.4 (18.9)* |
| Bilirubin (umol/L) | 41,895 (65.5) | 11.1 (6.9) | 11.1 (6.8)* |
| Full blood count | | | |
| Haemoglobin (g/L) | 46,129 (72.1) | 13.5 (1.6) | 13.5 (1.6)* |
| Lymphocytes (×10$^9$/L) | 42,502 (66.5) | 1.9 (0.7) | 1.9 (0.8)* |
| Mean cell volume (fL) | 45,010 (70.4) | 91.1 (6.2) | 91.1 (6.2)* |
| Monocytes (×10$^9$/L) | 42,607 (66.6) | 0.6 (0.4) | 0.6 (0.4)* |
| Neutrophils (×10$^9$/L) | 40,592 (63.5) | 4.6 (2.1) | 4.6 (2.1)* |
| Platelets (×10$^9$/L) | 45,239 (70.7) | 272.5 (88.5) | 272.2 (88.3)* |
| Total white cell count (×10$^9$/L) | 44,714 (69.9) | 7.3 (2.5) | 7.3 (2.5)* |
| Inflammatory markers | | | |
| C-reactive protein (mg/L) | 14,703 (23.0) | 13.3 (30.1) | 11.7 (27.3)* |
| Erythrocyte sedimentation rate (mm/h) | 22,931 (35.9) | 17.4 (21.4) | 17.6 (21.0)* |
| Cancer diagnosis within 6 months | | | |
| *Yes* | 908 (1.4) | | 1.4 |
| *No* | 63,065 (98.6) | | 98.6 |
| Total | **63,973** | | **50×63973** |

*Imputed data.

ALT, alanine aminotransferase; AST, aspartate aminotransferase; SD, standard deviation.

test results were dichotomised as abnormal/normal in each imputed dataset using standard laboratory ranges (S1 Table). Rubin's rules were used to combine results across the imputed datasets [16].

## Model derivation

Three prediction models were derived in the complete dataset: a symptoms-only model (Sm), a symptoms and tests model (STm), and a simple tests-only model (Tm). The *mim* Stata command was used to select variables for each model in the imputed data using backwards stepwise logistic regression, using a *p*-value of <0.01 for inclusion. Candidate variables for Sm included age-group, sex, smoking status, and clinical features found to be associated with a cancer diagnosis within 6 months in males and females (Table 2) [5]. Candidate variables for the STm also included the blood tests most commonly requested by GPs in patients with UWL and tests used in prognostic scores for patients with cancer (Table 3) [5,19,20]. For the Tm, candidate variables included age-group, sex, smoking status, and the blood tests, and as we intended to derive a parsimonious model, we chose the quantum over component tests; for example, the total white cell count was included rather than the white cell subtypes (Table 4). The most complex model (STm) had at least 15 events per variable [21].

**Table 2. Sm—Symptoms-only model.**

| Covariate | | Coefficient (95% CI) | Odds Ratio (95% CI) | p-value |
|---|---|---|---|---|
| *Demographics* | | | | |
| Age-group (ref = 60–69 years) | 18–39 | −3.85 (−4.75–2.96) | 0.02 (0.01–0.05) | <0.001 |
| | 40–49 | −1.93 (−2.40–1.46) | 0.14 (0.09–0.23) | <0.001 |
| | 50–59 | −1.11 (−1.43–0.78) | 0.33 (0.24–0.46) | <0.001 |
| | 70–79 | 0.63 (0.43–0.82) | 1.87 (1.53–2.28) | <0.001 |
| | 80+ | 0.72 (0.52–0.93) | 2.06 (1.69–2.52) | <0.001 |
| Sex (ref = female) | male | 0.77 (0.63–0.91) | 2.16 (1.87–2.48) | <0.001 |
| *Risk factors* | | | | |
| Smoking status (ref = non-smoker) | current | 0.61 (0.37–0.84) | 1.84 (1.45–2.32) | <0.001 |
| | ex-smoker | 0.21 (−0.02–0.43) | 1.23 (0.98–1.54) | 0.068 |
| *Symptoms* | | | | |
| Abdominal pain | Yes | 0.77 (0.56–0.98) | 2.17 (1.76–2.67) | <0.001 |
| Appetite loss | Yes | 0.61 (0.33–0.89) | 1.84 (1.39–2.43) | <0.001 |
| Dyspepsia | Yes | 0.46 (0.16–0.75) | 1.58 (1.18–2.12) | <0.001 |
| *Sign* | | | | |
| Abdominal mass | Yes | 1.09 (0.59–1.59) | 2.98 (1.81–4.93) | <0.001 |
| Chest signs | Yes | 1.32 (0.60–2.04) | 3.73 (1.81–7.68) | <0.001 |
| Iron deficiency anaemia | Yes | 1.15 (0.82–1.48) | 3.16 (2.27–4.41) | <0.001 |
| Jaundice | Yes | 1.89 (1.33–2.45) | 6.62 (3.77–11.6) | <0.001 |
| Lymphadenopathy | Yes | 1.54 (0.73–2.35) | 4.67 (2.08–10.5) | <0.001 |
| Venous thromboembolism | Yes | 1.10 (0.47–1.73) | 3.00 (1.60–5.64) | <0.001 |
| *Constant* | | −4.97 (−5.2–4.75) | | <0.001 |

Candidate covariates included in the backwards stepwise selection procedure using $p < 0.01$ to retain covariates included the following: sex, age-group, smoking status, abdominal mass, abdominal pain, appetite loss, back pain, chest pain (noncardiac), chest signs, change in bowel habit, dyspepsia, dysphagia, iron deficiency anaemia, jaundice, lymphadenopathy, and venous thromboembolism.

## Internal cross-validation

Ten-fold internal cross-validation was used to assess overall model performance using the mean predicted probability for each patient across all 50 imputation datasets with the *cvauroc* command in Stata [22]. Model performance was assessed using discrimination and calibration statistics. Discrimination was quantified using the area under the curve (AUC) and 95% confidence intervals calculated using bootstrap resampling. Calibration plots were generated using Stata's *pmcalplot* command to assess how the predicted probabilities derived by each model correspond to the observed proportion of patients diagnosed with cancer [23].

## Decision curve analysis

We then used decision curve analysis (DCA) to compare the standardised net benefit (SNB) and proportion of investigations avoided by the Sm, STm, and Tm with scenarios where no prediction model was used (i.e., treat everyone or treat nobody) across a range of risk thresholds (threshold probabilities) using the *dca* command in Stata [24]. In general, the strategy with the highest net benefit (the highest plotted curve) is considered to have the greatest clinical utility at any given risk threshold [25]. Net benefit represents the proportion of the studied population with true positive results minus the proportion with false positives multiplied by the odds of cancer at each risk threshold. To ease interpretation, we calculated the SNB to give the proportion of the maximum achievable utility attained by each model (SNB = NB /

**Table 3. STm—Symptoms and test model.**

| Covariate | | Coefficient (95% CI) | Odds ratio (95% CI) | p-value | Risk score* |
|---|---|---|---|---|---|
| Demographics | | | | | |
| Age-group (ref = 60–69 yrs) | 18–39 yrs | −3.34 (−4.24–2.44) | 0.04 (0.01–0.09) | <0.001 | −10 |
| | 40–49 yrs | −1.61 (−2.09–1.13) | 0.20 (0.12–0.32) | <0.001 | −5 |
| | 50–59 yrs | −0.96 (−1.3–0.63) | 0.38 (0.27–0.53) | <0.001 | −3 |
| | 70–79 yrs | 0.36 (0.16–0.57) | 1.44 (1.17–1.77) | 0.001 | 1 |
| | 80+ yrs | 0.33 (0.11–0.54) | 1.38 (1.12–1.72) | 0.003 | 1 |
| Sex (ref = female) | Male | 0.53 (0.38–0.69) | 1.70 (1.46–1.99) | <0.001 | 2 |
| Smoking status (ref = never) | Current | 0.60 (0.35–0.85) | 1.82 (1.41–2.34) | <0.001 | 2 |
| | Ex-smoker | 0.18 (−0.06–0.42) | 1.20 (0.94–1.53) | 0.144 | 1 |
| Symptoms | | | | | |
| Abdominal pain | Yes | 0.64 (0.42–0.86) | 1.90 (1.52–2.37) | <0.001 | 2 |
| Change in bowel habit | Yes | 0.71 (0.23–1.18) | 2.03 (1.26–3.26) | 0.004 | 2 |
| Dyspepsia | Yes | 0.47 (0.16–0.77) | 1.59 (1.17–2.17) | 0.003 | 1 |
| Signs | | | | | |
| Iron Deficiency Anaemia | Yes | 0.51 (0.15–0.88) | 1.67 (1.16–2.41) | 0.006 | 2 |
| Jaundice | Yes | 0.85 (0.24–1.45) | 2.33 (1.27–4.28) | 0.006 | 3 |
| Blood tests | | | | | |
| Low albumin | Yes | 0.36 (0.17–0.55) | 1.43 (1.18–1.73) | <0.001 | 1 |
| Raised alkaline phosphatase | Yes | 0.67 (0.48–0.86) | 1.95 (1.61–2.36) | <0.001 | 2 |
| Raised AST/ALT | Yes | 0.69 (0.39–0.99) | 1.99 (1.47–2.70) | <0.001 | 2 |
| Raised CRP | Yes | 1.41 (1.18–1.64) | 4.09 (3.24–5.16) | <0.001 | 4 |
| Low haemoglobin | Yes | 0.37 (0.19–0.55) | 1.45 (1.21–1.74) | <0.001 | 1 |
| Low mcv | Yes | −0.47 (−0.68–0.25) | 0.63 (0.50–0.78) | <0.001 | −1 |
| Raised platelets | Yes | 0.50 (0.31–0.70) | 1.65 (1.36–2.02) | <0.001 | 2 |
| Raised monocytes | Yes | 0.68 (0.50–0.86) | 1.97 (1.65–2.36) | <0.001 | 2 |
| Low lymphocytes | Yes | 0.36 (0.18–0.53) | 1.43 (1.20–1.70) | <0.001 | 1 |
| Constant | | −6.09 (−6.36–5.81) | | <0.001 | |

Candidate covariates included in the backwards stepwise selection procedure using $p < 0.01$ to retain covariates included the following: sex, age-group, smoking status, abdominal mass, abdominal pain, appetite loss, back pain, chest pain (noncardiac), chest signs, change in bowel habit, dyspepsia, dysphagia, iron deficiency anaemia, jaundice, lymphadenopathy, venous thromboembolism, albumin, alkaline phosphatase, bilirubin, C-reactive protein, erythrocyte sedimentation rate, haemoglobin, liver enzymes (AST/ALT), lymphocytes, mean cell volume, monocytes, neutrophils, and platelets.

*Conversion factor = 0.32. The conversion factor is used to translate coefficients into the risk score, rounded to the nearest integer.

ALT, alanine aminotransferase; AST, aspartate aminotransferase; CRP, C-reactive protein; mcv, mean cell volume.

prevalence of cancer) [26]. An alternative presentation of DCA is the proportion of patients who would avoid further investigation without missing a cancer diagnosis at each risk threshold [27].

## Clinical risk scores

Finally, to demonstrate how these models could be used in clinical practice, we followed established methods to develop 2 simple clinical risk scores for the STm and Tm [28]. The risk score associated with each variable was derived by multiplying each coefficient by the same conversion factor and rounding the result to the nearest whole number. We calculated the mean point score for each patient across the imputation datasets and constructed a 2 × 2 table using each total score as the cutoff. We calculated the sensitivity, specificity, positive likelihood ratio (PLR), negative likelihood ratio (NLR), positive predictive value (PPV), and negative predictive values (NPVs) for each score.

**Table 4. Tm—Tests-only model.**

| Covariate | | Coefficient (95% CI) | Odds ratio (95% CI) | *p*-value | Risk score* |
|---|---|---|---|---|---|
| *Demographics* | | | | | |
| Age-group (ref = 60–69 years) | 18–39 | −3.31 (−4.20–2.41) | 0.04 (0.01–0.09) | <0.001 | −10 |
| | 40–49 | −1.61 (−2.08–1.14) | 0.20 (0.12–0.32) | <0.001 | −5 |
| | 50–59 | −0.94 (−1.28–0.61) | 0.39 (0.28–0.54) | <0.001 | −3 |
| | 70–79 | 0.36 (0.15–0.56) | 1.43 (1.17–1.75) | 0.001 | 1 |
| | 80+ | 0.26 (0.06–0.47) | 1.30 (1.06–1.60) | 0.012 | 1 |
| Sex (ref = female) | male | 0.61 (0.46–0.76) | 1.84 (1.59–2.14) | <0.001 | 2 |
| *Blood tests* | | | | | |
| Low albumin | Yes | 0.38 (0.19–0.57) | 1.47 (1.21–1.78) | <0.001 | 1 |
| Raised alkaline phosphatase | Yes | 0.70 (0.51–0.89) | 2.01 (1.66–2.43) | <0.001 | 2 |
| Raised AST/ALT | Yes | 0.70 (0.41–1.00) | 2.02 (1.51–2.71) | <0.001 | 2 |
| Raised CRP (>10) | Yes | 1.50 (1.28–1.73) | 4.50 (3.60–5.64) | <0.001 | 5 |
| Raised total white cell count | Yes | 0.60 (0.40–0.79) | 1.82 (1.50–2.20) | <0.001 | 2 |
| Raised platelets | Yes | 0.52 (0.32–0.72) | 1.68 (1.38–2.04) | <0.001 | 2 |
| Low haemoglobin | Yes | 0.44 (0.26–0.61) | 1.55 (1.30–1.85) | <0.001 | 1 |
| Constant | | −5.69 (−5.93–5.46) | | <0.001 | |

Candidate covariates included in the backwards stepwise selection procedure using $p < 0.01$ to retain covariates included the following: sex, age-group, smoking status, albumin, alkaline phosphatase, bilirubin, CRP, erythrocyte sedimentation rate, haemoglobin, liver enzymes (AST/ALT), platelets, and total white cell count.

*Conversion factor = 0.32. The conversion factor is used to translate coefficients into the risk score, rounded to the nearest integer.

ALT, alanine aminotransferase; AST, aspartate aminotransferase; CRP, C-reactive protein.

## Sensitivity analysis

We refitted the models using the missing indicator method to assess our approach to multiple imputation. We refitted the models using continuous blood test results to explore the impact on discrimination and calibration statistics. We used Stata's *mfpmi* command to select the most appropriate functional form for each continuous covariate in relation to the outcome [29].

# Results

## Study participants

In the derivation cohort of 63,973 adults aged ≥18 years with UWL recorded, 908 (1.4%) were diagnosed with cancer within 6 months of the index date, of whom 902 (99.3%) were aged ≥40 years. Table 1 summarises the baseline characteristics of the study population. Patients with UWL were more commonly females (58.2%), aged ≥60 years (51.8%), and of normal body mass index (52.9%). The most commonly recorded clinical features were abdominal pain (5.9%), back pain (5.1%), noncardiac chest pain (2.9%), and dyspepsia (2.8%) (Table 1). The most commonly recorded tests were haemoglobin (72.1%), platelets (70.7%), and total white cell count (69.9%).

## Missing data

A total of 32,723 (51.15%) patients had missing data on smoking status, 36,356 (56.8%) on alcohol consumption, and 7,235 (11.3%) had no body mass index recorded (Table 1). The most commonly missing blood tests were liver enzymes (53,062, 82.9%), C-reactive protein (49,270, 77.0%), and erythrocyte sedimentation rate (41,042, 64.1%). Imputation diagnostics were deemed satisfactory for all imputed variables (S1 Fig). The direction of the estimates was

the same, and the confidence intervals overlapping when comparing variables included in the final imputed and missing indicator models.

## Model development

In the final Sm, 12 of 17 candidate variables were associated with cancer (Table 2), of which concurrent jaundice (adjusted odds ratio 6.62 (95% CI 3.77 to 11.63)) and lymphadenopathy (4.67 (2.08 to 10.47)) were most predictive. Out of 29 candidate variables, 17 were retained in the final STm (Table 3), of which a raised C-reactive protein (4.09 (3.24 to 5.16) and concurrent jaundice (2.33 (1.27 to 4.28)) were most predictive. Out of 12 candidate predictor variables, 9 were retained in the final Tm (Table 4), of which raised C-reactive protein (4.50 (3.60 to 5.64)) and raised liver enzymes (2.02 (1.51 to 2.71)) were most predictive.

## Internal validation

The AUC for both the STm (0.92 (0.91 to 0.93)) and the Tm (0.91 (0.90 to 0.92)) showed discrimination, which was superior to the Sm (0.79 (0.78 to 0.81)) (Table 5). However, the calibration statistics showed that the Sm was better calibrated compared to the STm and Tm. The calibration plots showed that the difference in calibration statistics was mainly due to underprediction in the highest decile of risk for the STm and Tm that was not seen for the Sm (Fig 1). Refitting the STm and Tm with continuous blood test results instead of dichotomised blood test results made negligible difference to model performance (Table 5, S2 Fig, S2 Table).

## Decision curve analysis

The STm had greatest clinical utility (Fig 2A). The STm had higher SNB than the Sm for the risk thresholds of 0.4% to 18%, and the Tm had greater net benefit to the Sm for 0.5% to 15% (Fig 2A, S3 Table). At a cancer risk threshold of 1%, these differences translate into a 55% reduction in further cancer testing if using the STm compared to investigating all patients, a 19% reduction compared to using the Sm, or a 2% reduction compared to the Tm (Fig 2B, S3 Table).

## Examples of applying the clinical risk scores

Figs 3 and 4 show diagnostic accuracy statistics corresponding to each possible point score for the STm and the Tm, respectively. S4 and S5 Tables show how these statistics apply to 100,000 patients with UWL for the STm and the Tm risk score thresholds, respectively. Box 1 gives

**Table 5. Prediction model discrimination and calibration statistics.**

| Models | Discrimination | Calibration | | |
|---|---|---|---|---|
| | AUC (95% CI) | E:O | CITL | Slope |
| Sm | 0.79 (0.78–0.81) | 0.995 | 0.005 | 0.993 |
| STm (dichotomous primary analysis) | 0.92 (0.91–0.93) | 0.825 | 0.210 | 1.197 |
| *STm (continuous sensitivity analysis)* | *0.92 (0.91–0.93)* | *0.820* | *0.220* | *1.199* |
| Tm (dichotomous primary analysis) | 0.91 (0.90–0.92) | 0.824 | 0.209 | 1.213 |
| *Tm (continuous sensitivity analysis)* | *0.92 (0.91–0.93)* | *0.826* | *0.211* | *1.199* |

AUC, area under the curve; CITL, calibration in the large; E:O, ratio of expected (predicted) probability vs observed frequency of the outcome; Sm, symptoms-only model; STm, symptoms and tests model; Tm, tests-only model.

An AUC of 0.5 represents chance, and 1 represents perfect ability to discriminate between patients who will and patients who will not be diagnosed with cancer [47].

Perfect calibration has a calibration slope of 1, a CITL of 0, and an O:E ratio of 1 [41].

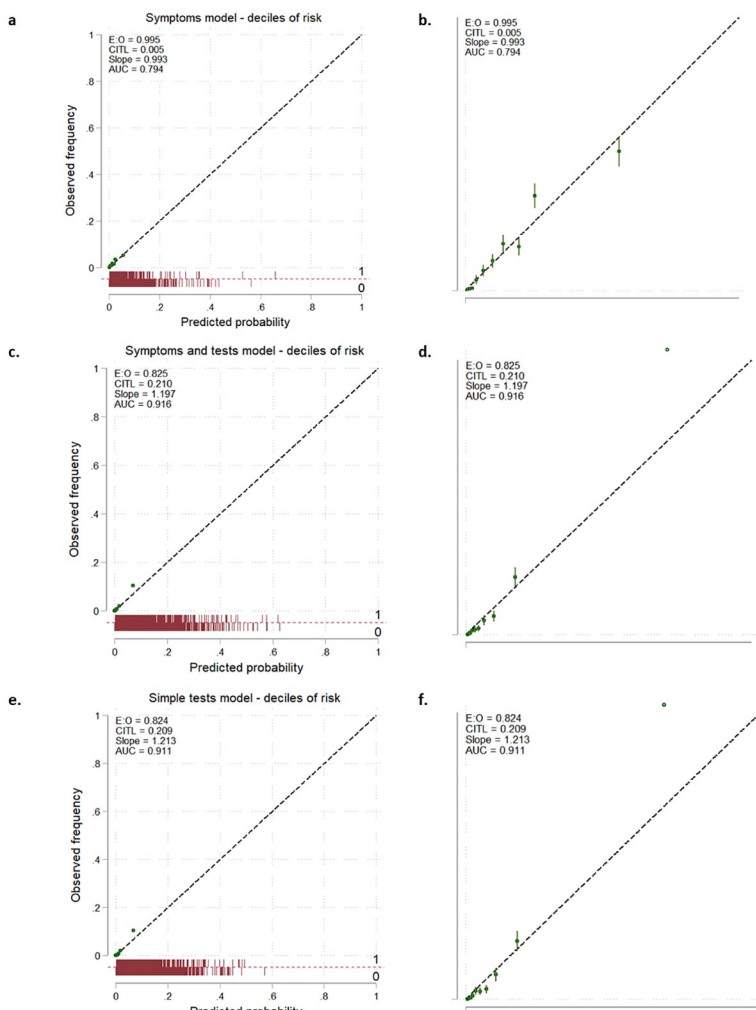

**Fig 1. Calibration plots for the Sm (1a and 1b), the STm (1c and 1d), and the Tm (1e and 1f).** Green points are deciles of predicted probability with error bars. The right hand panels (1b, 1d, and 1e) are zoomed in to show in detail the first 0.1% of predicted probability and observed frequency. AUC, area under the curve; CITL, calibration in the large; E:O, ratio of expected (predicted) probability vs observed frequency of the outcome; Sm, symptoms-only model; STm, symptoms and tests model; Tm, tests-only model.

examples of how the STm score might be used in clinical practice, for example, by choosing the optimal threshold to sufficiently rule-out further cancer investigation.

## Discussion

### Summary of findings

Combinations of multiple simple test results were discriminative between patients with and without cancer, were well calibrated at the levels of risk that decisions to investigate are made in primary care, and showed superior clinical utility when compared to symptoms and signs. We present stand-alone risk scores that could be used by GPs to guide test selection and interpretation. The simplest includes age-group, sex, and 7 primary care blood tests (alkaline phosphatase, liver enzymes, albumin, C-reactive protein, haemoglobin, platelets, and total white cell count). They could be used to select patients with UWL who do not warrant further cancer investigation in addition to those that do.

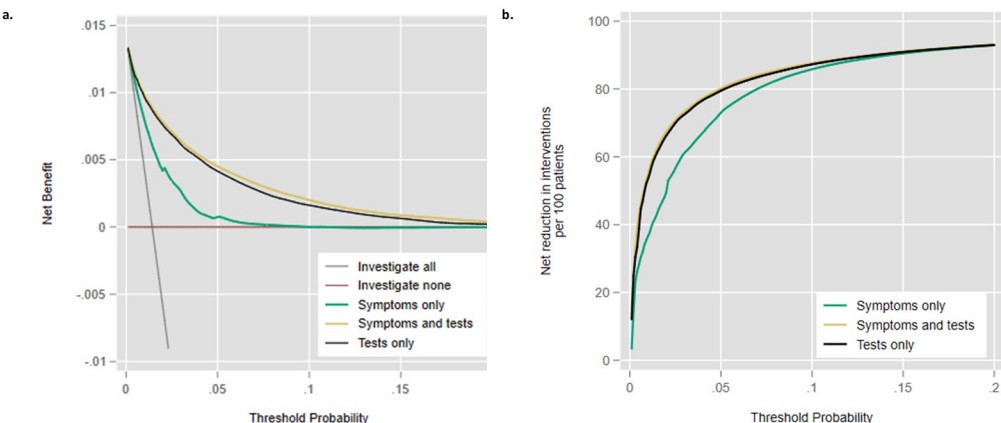

**Fig 2.** Decision curve analysis comparing the three models in terms of net benefit (Fig 2a) and investigations avoided (Fig 2b). DCA, decision curve analysis.

## Strengths and limitations

Our study design aimed to minimise bias. We excluded patients with objective evidence of intentional weight loss, restricted co-occurring clinical features to the time of the UWL presentation [5], and included only the first UWL code [2,30]. We were reliant on electronic health record (EHR) codes to define UWL as weight is not recorded frequently enough [13]. It is unclear how recording bias relates to coding for UWL, which occurs when GPs preferentially code clinical features that they associate with cancer and can lead to inflated estimates of association for these features [31]. We excluded patients with a past history of cancer and only

| Score | TP | FP | FN | TN | Sn (%) | Sp (%) | PLR | NLR | PPV (%) | NPV (%) |
|---|---|---|---|---|---|---|---|---|---|---|
| -11 | 908 | 63065 | 0 | 0 | 100 | N/E | N/E | N/E | 1.42 | 100 |
| -10 | 908 | 63011 | 0 | 54 | 100 | 0.09 | 1.00 | 0 | 1.42 | 100 |
| -9 | 908 | 61640 | 0 | 1425 | 100 | 2.26 | 1.02 | 0 | 1.45 | 100 |
| -8 | 908 | 58467 | 0 | 4598 | 100 | 7.29 | 1.08 | 0 | 1.53 | 100 |
| -7 | 908 | 55344 | 0 | 7721 | 100 | 12.2 | 1.14 | 0 | 1.61 | 100 |
| -6 | 908 | 53119 | 0 | 9946 | 100 | 15.8 | 1.19 | 0 | 1.68 | 100 |
| -5 | 908 | 50735 | 0 | 12330 | 100 | 19.6 | 1.24 | 0 | 1.76 | 100 |
| -4 | 907 | 49186 | 1 | 13879 | 99.9 | 22.0 | 1.28 | 0 | 1.81 | 100 |
| -3 | 906 | 47187 | 2 | 15878 | 99.8 | 25.2 | 1.33 | 0.01 | 1.88 | 100 |
| -2 | 904 | 44756 | 4 | 18309 | 99.6 | 29.0 | 1.40 | 0.02 | 1.98 | 100 |
| -1 | 902 | 41821 | 6 | 21244 | 99.3 | 33.7 | 1.50 | 0.02 | 2.11 | 100 |
| 0 | 900 | 38919 | 8 | 24146 | 99.1 | 38.3 | 1.61 | 0.02 | 2.26 | 100 |
| 1 | 895 | 35673 | 13 | 27392 | 98.6 | 43.4 | 1.74 | 0.03 | 2.45 | 100 |
| 2 | 889 | 30976 | 19 | 30089 | 97.9 | 50.9 | 1.99 | 0.04 | 2.79 | 99.9 |
| 3 | 875 | 25753 | 33 | 37312 | 96.4 | 59.2 | 2.36 | 0.06 | 3.29 | 99.9 |
| 4 | 853 | 19938 | 55 | 43127 | 93.9 | 68.4 | 2.97 | 0.09 | 4.10 | 99.9 |
| 5 | 822 | 14710 | 86 | 48355 | 90.5 | 76.7 | 3.88 | 0.12 | 5.29 | 99.8 |
| 6 | 784 | 10706 | 124 | 52359 | 86.3 | 83.0 | 5.09 | 0.16 | 6.82 | 99.8 |
| 7 | 729 | 7439 | 179 | 55626 | 80.3 | 88.2 | 6.81 | 0.22 | 8.93 | 99.7 |
| 8 | 667 | 5207 | 241 | 57858 | 73.5 | 91.7 | 8.90 | 0.29 | 11.4 | 99.6 |
| 9 | 583 | 3654 | 325 | 59411 | 64.2 | 94.2 | 11.1 | 0.38 | 13.8 | 99.5 |
| 10 | 487 | 2475 | 421 | 60590 | 53.6 | 96.1 | 13.7 | 0.48 | 16.4 | 99.3 |
| 11 | 389 | 1591 | 519 | 61474 | 42.8 | 97.5 | 17.0 | 0.59 | 19.6 | 99.2 |
| 12 | 262 | 996 | 646 | 62069 | 28.9 | 98.4 | 18.3 | 0.72 | 20.8 | 99.0 |
| 13 | 186 | 575 | 722 | 62490 | 20.5 | 99.1 | 22.5 | 0.80 | 24.4 | 98.9 |
| 14 | 125 | 324 | 783 | 62741 | 13.8 | 99.5 | 26.8 | 0.87 | 27.8 | 98.8 |
| 15 | 71 | 159 | 837 | 62906 | 7.82 | 99.7 | 31.0 | 0.92 | 30.9 | 98.7 |
| 16 | 45 | 77 | 863 | 62988 | 4.96 | 99.9 | 40.6 | 0.95 | 36.9 | 98.6 |
| 17 | 19 | 27 | 889 | 63038 | 2.09 | 100 | 48.9 | 0.98 | 41.3 | 98.6 |
| 18+ | 10 | 17 | 898 | 63048 | 1.10 | 100 | 40.9 | 0.99 | 37.0 | 98.6 |

**Fig 3. Full breakdown of STm model performance.** FN, false negative; FP, false negative; N/E, not estimable; NLR, negative likelihood ratio; NPV, negative predictive value; PLR, positive likelihood ratio; PPV, positive predictive value; Sn, sensitivity; Sp, specificity; STm, symptoms and tests model; TN, true negative; TP, true positive.

| Score | TP | FP | FN | TN | Sn (%) | Sp (%) | PLR | NLR | PPV (%) | NPV (%) |
|---|---|---|---|---|---|---|---|---|---|---|
| -10 | 908 | 63065 | 0 | 0 | 100 | N/E | N/E | N/E | 1.42 | 100 |
| -9 | 908 | 57861 | 0 | 5204 | 100 | 8.25 | 1.09 | 0 | 1.55 | 100 |
| -8 | 908 | 55062 | 0 | 8003 | 100 | 12.7 | 1.15 | 0 | 1.62 | 100 |
| -7 | 908 | 51801 | 0 | 11264 | 100 | 17.9 | 1.22 | 0 | 1.72 | 100 |
| -6 | 908 | 49986 | 0 | 13079 | 100 | 20.7 | 1.26 | 0 | 1.78 | 100 |
| -5 | 907 | 49539 | 1 | 13526 | 99.9 | 21.4 | 1.27 | 0.01 | 1.80 | 100 |
| -4 | 905 | 46990 | 3 | 16075 | 99.7 | 25.5 | 1.34 | 0.01 | 1.89 | 100 |
| -3 | 905 | 45476 | 3 | 17589 | 99.7 | 27.9 | 1.38 | 0.01 | 1.95 | 100 |
| -2 | 899 | 41089 | 9 | 21976 | 99.0 | 34.8 | 1.52 | 0.03 | 2.14 | 100 |
| -1 | 896 | 38674 | 12 | 24391 | 98.7 | 38.7 | 1.61 | 0.03 | 2.26 | 100 |
| 0 | 895 | 35884 | 13 | 27181 | 98.6 | 43.1 | 1.73 | 0.03 | 2.43 | 100 |
| 1 | 885 | 32309 | 23 | 30756 | 97.5 | 48.8 | 1.90 | 0.05 | 2.67 | 99.9 |
| 2 | 861 | 25792 | 47 | 37273 | 94.8 | 59.1 | 2.32 | 0.09 | 3.23 | 99.9 |
| 3 | 844 | 20112 | 64 | 42953 | 93.0 | 68.1 | 2.91 | 0.10 | 4.03 | 99.9 |
| 4 | 805 | 13840 | 103 | 49225 | 88.7 | 78.1 | 4.04 | 0.15 | 5.50 | 99.8 |
| 5 | 763 | 9879 | 145 | 53186 | 84.0 | 84.3 | 5.36 | 0.19 | 7.17 | 99.7 |
| 6 | 708 | 6882 | 200 | 56183 | 78.0 | 89.1 | 7.15 | 0.25 | 9.33 | 99.6 |
| 7 | 638 | 4985 | 270 | 58080 | 70.3 | 92.1 | 8.89 | 0.32 | 11.3 | 99.5 |
| 8 | 555 | 3649 | 353 | 59416 | 61.1 | 94.2 | 10.6 | 0.41 | 13.2 | 99.4 |
| 9 | 450 | 2618 | 458 | 60447 | 49.6 | 95.8 | 11.9 | 0.53 | 14.7 | 99.2 |
| 10 | 351 | 1755 | 557 | 61310 | 38.7 | 97.2 | 13.9 | 0.63 | 16.7 | 99.1 |
| 11 | 239 | 1143 | 669 | 61922 | 26.3 | 98.2 | 14.5 | 0.75 | 17.3 | 98.9 |
| 12 | 170 | 721 | 738 | 62344 | 18.7 | 98.9 | 16.4 | 0.82 | 19.1 | 98.8 |
| 13 | 111 | 419 | 797 | 62646 | 12.2 | 99.3 | 18.4 | 0.88 | 20.9 | 98.7 |
| 14 | 58 | 205 | 850 | 62860 | 6.39 | 99.7 | 19.7 | 0.94 | 22.1 | 98.7 |
| 15 | 29 | 93 | 879 | 62972 | 3.19 | 99.9 | 21.7 | 0.97 | 23.8 | 98.6 |
| 16 | 15 | 38 | 893 | 63027 | 1.65 | 99.9 | 27.4 | 0.98 | 28.3 | 98.6 |
| 17 | 7 | 7 | 901 | 63058 | 0.77 | 100 | 69.5 | 0.99 | 50.0 | 98.6 |
| 18 | 2 | 0 | 906 | 63065 | 0.22 | 100 | N/E | 1.00 | 100 | 98.6 |

**Fig 4. Full breakdown of Tm model performance.** FN, xxxx; FP, xxxx; N/E, xxxx; NLR, negative likelihood ratio; NPV, negative predictive value; PLR, positive likelihood ratio; PPV, positive predictive value; Sn, xxxx; Sp, xxxx; Tm, tests-only model; TN, xxxx; TP, xxxx.

included cancers coded within 6 months of the UWL date to ensure that we investigated a first diagnosis of a cancer associated with UWL.

By utilising multiple imputation to replace missing risk factor and continuous test result data, we could produce precise estimates for combinations of multiple covariates. Previous studies have not done this and have had to focus on single or pairs of blood test abnormalities [3]. We included auxiliary variables to increase the likelihood that missing values could be accurately predicted by the observed data (that they are missing at random) [16]. However, there is no established method to formally evaluate whether this was successful. Imputation allowed us to combine multiple blood test results and to show that once blood tests are modelled with sex and age, there appears to be no need to include additional risk factors and clinical features.

We dichotomised each blood test for the primary analysis to derive a simple risk score for use in clinical practice using the upper or lower boundary of the normal reference range. This can have limitations. Firstly, by dichotomising a continuous variable, information is lost by grouping slightly and extremely abnormal results together. Secondly, choosing raised values to define abnormal might be unhelpful for cancer sites associated with low values (and vice versa). Thirdly, we chose the upper limit of the normal range to define abnormal for blood tests where there is no consensus on how to define abnormal. Refitting the models to include continuous linear and fractional polynomial terms made no meaningful difference to model performance.

We required a testing strategy appropriate for a composite of all cancer types. The literature reports that the direction of blood test abnormalities are similar for most cancers and that a pro-inflammatory state underpins many cancers and cancer cachexia [19,20,32–39]. While this supports our approach, it remains likely the composite cancer outcome is partly

responsible for the underprediction observed at the highest decile of risk. It is unlikely that this would have adverse clinical consequences because GPs' decision to refer for invasive testing is likely to be triggered at lower thresholds than this.

We used 10-fold internal cross-validation to derive estimates of predictive performance [40]. However, internal validation may produce overoptimistic estimates and so external validation remains necessary to assess the generalisability of our findings in settings, populations, and subgroups of interest [41,42]. Primary care data will be identified from alternative clinical systems for the same time period or from the same clinical systems for a later time period to account for variation in UK practice, international settings with alternative approached to weight measurement and weight loss recording and where a different spectrum of patients consults with primary care, and in systems where similar blood tests are used with differing degrees of missingness.

## Findings in context

One previous study developed a risk score to predict cancer in a cohort of 256 patients referred for the investigation of UWL (AUC 0.90 (95% CI 0.88 to 0.92)) including the following: age ≥80 years, white blood cells, albumin, alkaline phosphatase, and lactate dehydrogenase [10]. The AUC was notably lower when externally validated (0.70 (0.61 to 0.78)) in a cohort of 290 consecutive patients referred to hospital with UWL [9]. This study also developed a simpler 3-variable score that included age, alkaline phosphatase, and albumin and gave an AUC of 0.74 (0.66 to 0.81) [9]. The models we developed here produced higher AUCs than these models and, more importantly, include a cohort of all patients presenting to primary care with UWL, not those referred to hospital for further investigation of UWL.

Two existing primary care prediction models incorporate multiple symptoms and risk factors to estimate cancer risk over a 2-year period for 1.24 million females (AUC 0.85 (95% CI 0.84 to 0.85)) and 1.26 million males (0.87 (0.88 to 0.89)) aged 25 to 89 years [43,44]. They also demonstrate good calibration at the lower deciles of risk and miscalibration at the highest decile of risk. The relative timing of symptoms was not reported. Blood tests were not included, except haemoglobin results in the 12 months before to 2 months after study entry were used to define anaemia as a baseline risk factor. Consequently, the design and reporting of these models make it impossible to understand the diagnostic value of symptoms and blood tests co-occurring with UWL.

## Implications for research and clinical practice

DCA allowed us to demonstrate the importance of simple tests in comparison to symptoms and signs. It was used to define how risk thresholds related to the probable "yield" of referrals in diagnosing cancer through hospital-based invasive investigation [25] However, DCA cannot itself define the acceptable balance between the number of people referred to each person with cancer referred or between the number of people spared referral to each person with cancer not referred These trade-offs are for patients, clinicians, and society at large to decide and could be evaluated further in health economic analysis. The examples shown in Box 1 illustrates 4 examples of how a risk score could inform these decisions. Moreover, these examples could help GPs develop watchful waiting strategies for patients with intermediate risks of having cancer, perhaps scheduling periodic review and blood test reevaluation to examine whether the risk has changed.

There is a dearth of research on how risk scores for cancer are best adopted into clinical practice. We intended to derive an intuitive risk score that mirrors clinical practice by focussing on simple combinations of clinical features including commonly used and available blood

## Box 1. Consequences of applying the STm score in clinical practice (Tables 3 and S4)

### Clinical example

A 52-year-old woman with UWL, no other clinical features, a low albumin, high alkaline phosphatase, and a raised C-reactive protein corresponds to an STm score of 4: sensitivity 93.9%, specificity 68.4%. At this threshold, 23 people would be referred for each person with cancer referred, and 784 people would be spared investigation for each person with cancer not referred. Per 100,000 people with UWL, 1,333 with cancer would be referred, 31,152 would be referred unnecessarily, 67,429 correctly spared referral, and 86 people with cancer would not be referred.

### Example of maximising sensitivity to rule-out cancer

An STm score of 1 prioritises ruling-out cancer by maximising sensitivity: sensitivity 98.6%, specificity 43.4%. At this threshold, 40 people with UWL would be referred for each person with cancer referred, and 2,139 people with UWL spared referral for each person with cancer not referred. Per 100,000 people with UWL, 1,399 people with cancer would be referred, 55,797 people would be unnecessarily referred, 42,784 correctly spared referral, and 20 people with cancer not referred.

### Example of a threshold to balance ruling-in and ruling-out cancer

An STm score of 6 would balance ruling-in (PLR 5.09) and ruling-out (NLR 0.16) the need for referral: sensitivity 86.3%, specificity 83.0%. At this threshold, there would be 14 people referred for investigation for each person with cancer referred, and 422 would be spared investigation for each person with cancer not referred. Per 100,000 people with UWL, 1,225 people with cancer would be referred, 16,759 patients would be unnecessarily referred, 81,822 correctly spared referral, and 194 people with cancer not referred.

### Example of a threshold close to the NICE PPV threshold of 3%

An STm score of 2 is the closest to a PPV to 3%, the threshold chosen by NICE to warrant further investigation: sensitivity 97.9%, specificity 50.9%. At this threshold, 35 people would be referred for each person with cancer referred, and 1,730 patients would be spared investigation for each person with cancer not referred. Per 100,000 patients with UWL, 1,390 people with cancer would be referred, 48,403 patients would be unnecessarily referred, 50,178 correctly spared referral, and 29 people with cancer not referred.

tests. Each score could be completed by GPs by hand, used as an online calculator, or integrated as into the EHR. However, the limited literature shows that risk scores are underused, and GPs find predictions difficult to interpret or are distrustful of them, especially when they conflict with intuitive clinical judgement [45,46]. Further research is therefore required to understand their uptake and to assess whether their use impacts on cancer outcomes.

## Conclusions

Our findings suggest that combinations of simple blood test abnormalities could be used to identify patients with UWL who warrant referral for investigation, while people with combinations of normal results could be exempted from referral.

## Disclaimer

The lead author affirms that this manuscript is an honest, accurate, and transparent account of the study being reported; that no important aspects of the study have been omitted; and that any discrepancies from the study as planned (and, if relevant, registered) have been explained.

## Supporting information

**S1 TRIPOD Checklist.**
(DOCX)

**S1 Fig. Multiple imputation diagnostics for the first 20 imputations for continuous blood test variables.**
(DOCX)

**S2 Fig. Cross-validation plots for continuous STm model and the Tm-only model.** Green points are deciles of predicted probability with error bars. The right hand panels are zoomed in to show in detail the first 0.1% of predicted probability and observed frequency. AUC, area under the curve; CITL, calibration in the large; E:O, ratio of expected (predicted) probability vs observed frequency of the outcome; STm, symptoms and tests model; Tm, tests-only model.
(DOCX)

**S1 Table. Common laboratory test ranges in general practice.**
(DOCX)

**S2 Table. Full specification of the continuous STm and the Tm. STm, symptoms and test model; Tm, tests-only model.**
(DOCX)

**S3 Table. SNB and reduction in further investigation comparing models and investigating all patients or none.** SNB = net benefit / prevalence. SNB, standardised net benefit; STm, symptoms and test model; Sm, symptoms-only model; Tm, tests-only model.
(DOCX)

**S4 Table. STm risk score thresholds expressed per 100,000 patients with UWL investigated. STm, symptoms and test model; UWL, unexpected weight loss.**
(DOCX)

**S5 Table. Tm risk score thresholds expressed per 100,000 patients with UWL investigated. Tm, tests-only model; UWL, unexpected weight loss.**
(DOCX)

**S1 Text. Auxiliary variables used in the multiple imputation model.**
(DOCX)

## Author Contributions

**Conceptualization:** Brian D. Nicholson, Paul Aveyard, Constantinos Koshiaris, Rafael Perera, Willie Hamilton, Jason Oke, F. D. Richard Hobbs.

**Data curation:** Brian D. Nicholson.

**Formal analysis:** Brian D. Nicholson.

**Funding acquisition:** Brian D. Nicholson.

**Investigation:** Brian D. Nicholson.

**Methodology:** Brian D. Nicholson, Constantinos Koshiaris, Rafael Perera, Jason Oke.

**Supervision:** Paul Aveyard, Willie Hamilton, Jason Oke, F. D. Richard Hobbs.

**Writing – original draft:** Brian D. Nicholson.

**Writing – review & editing:** Brian D. Nicholson, Paul Aveyard, Constantinos Koshiaris, Rafael Perera, Willie Hamilton, Jason Oke, F. D. Richard Hobbs.

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
