## [Editor Report · Decision Letter 0]

19 Mar 2021

Dear Dr Nicholson, 

Thank you for submitting your manuscript entitled "Combinations of simple blood tests to identify primary care patients with unexpected weight loss for cancer investigation." for consideration by PLOS Medicine for our upcoming Special Issue.

Your manuscript has now been evaluated by the PLOS Medicine editorial staff and by the Guest Editors, and I am writing to let you know that we would like to send your submission out for external assessment.

Once your full submission is complete, your paper will undergo a series of checks in preparation for external assessment. 

Kind regards,

Richard Turner, PhD

rturner@plos.org

---

## [Decision Letter · Decision Letter 1]

21 Apr 2021

Dear Dr. Nicholson,

Thank you very much for submitting your manuscript "Combinations of simple blood tests to identify primary care patients with unexpected weight loss for cancer investigation." (PMEDICINE-D-21-01229R1) for consideration at PLOS Medicine. 

Your paper was discussed among the editors and evaluated by the Guest Editors for the Special issue. Comments from independent reviewers, including a statistical reviewer, are appended at the bottom of this email and any accompanying reviewer attachments can be seen via the link below:

[LINK]

In light of these reviews, we will not be able to accept the manuscript for publication in the journal in its current form, but we would like to invite you to submit a revised version that addresses the reviewers' and editors' comments fully. You will appreciate that we cannot make a decision about publication until we have seen the revised manuscript and your response, and we expect to seek re-review by one or more of the reviewers. 

We hope to receive your revised manuscript by May 12 2021 11:59PM. Please email us (plosmedicine@plos.org) if you have any questions or concerns.

Please let me know if you have any questions, and we look forward to receiving your revised manuscript. 

Sincerely,

Richard Turner, PhD

rturner@plos.org

We believe that the case for publication in PLOS Medicine would be enhanced if you were able to include an element of external validation, as suggested by one referee. 

To your data statement, please add contact information for those wishing to access data from CPRD.

To your title, please add a study descriptor following a colon, e.g., "...: A diagnostic accuracy study".

In the "Methods and findings" subsection of your abstract, please quote summary demographic details for study participants, and mention the recruitment period. 

Please add a new final sentence to the "Methods and findings" subsection of your abstract, which should begin "Study limitations include ..." or similar and should quote 2-3 of the study's main limitations. 

We ask you to revisit the wording of the "Conclusions" subsection of your abstract, and similar statements elsewhere in the paper, e.g., at the end of the main text, which seem to us somewhat too strong. We suggest "Our findings suggest that combinations ... could be used ..." or similar. 

After the abstract, please add a new and accessible "Author summary" section in non-identical prose. You may find it helpful to consult one or two recent research papers published in PLOS Medicine to get a sense of the preferred style. 

Please avoid the phrase "discriminated excellently". 

Please substitute "sex" for "gender" where appropriate, e.g., in table 1.

Throughout the text, please format reference call-outs as follows: "... clinical practice [8,9]." (noting the absence of spaces within the square brackets). 

Please remove the information on funding and competing interests from the end of the main text. In the event of publication, this will appear in the article metadata, via entries in the submission form. 

Please move the information on ethics approval from the end of the main text to the Methods section. 

In the reference list, please ensure that journal names are abbreviated correctly and consistently (e.g., "PLoS ONE", "PLoS Med."). 

Noting references 6 & 27, please ensure that all references have full access details. 

Please include the study protocol or prespecified analysis plan as a supplementary file, referred to in your Methods section. Please highlight analyses that were not prespecified. 

Please include a completed checklist for the most appropriate reporting guideline, e.g., TRIPOD, labelled "S1_TRIPOD_Checklist" or similar and referred to as such in the Methods section of your main text. 

In the checklist, please refer to individual items by section (e.g., "Methods") and paragraph number rather than by line or page numbers, as the latter generally change in the event of publication. 

Comments from the reviewers:

*** Reviewer #1: 

This study aims to derive and internally validate prediction models using co-occurring risk factors, symptoms, signs and blood test data to identify clinical features that could be used together to stratify cancer risk in patients attending primary care with UWL.

Comments:

"The protocol has been published" and "We followed the TRIPOD and RECORD reporting guidelines".

Can the authors please supply the protocol and reporting checklists in the supplementary material?

"Multiple imputation was used to replace missing values for smoking status, alcohol intake, body mass index, and blood tests using the mi suite of commands in Stata (15, 16). Fifty imputed datasets were created. "

Did the authors undertake any sensitivity analyses for their missing value approach? 

"Three prediction models were derived in the complete dataset: a symptoms only model (Sm), a symptoms and tests model (STm), and a simple tests only model (Tm). The mim Stata command was used to select variables for each model in the imputed data using backwards stepwise logistic regression, using a p-value of <0.01 for inclusion."

A technically appropriate statistical approach has been applied.

Internal cross-validation and conducting sensitivity analyses refitting models using continuous data, are rigorous and robust methodologies.

Overall, this is a clearly laid out and well written study. The authors have suitably acknowledged the main limitations, and present a thorough range of analyses in the supplementary material.

*** Reviewer #2: 

It is important to develop predictive tools that can help clinicians select patients with unexpected weight loss for further testing. I applaud the authors for conducting this important work. I however have some comments and concerns: 

1. How did you define UWL? In the first paragraph of the Methods section you write: "…12 months of data before their first UWL code (the "index date").". What is a "UWL code"? Some ICD-code? I suggest you specify. 

2. For the outcome, you only consider cancer diagnosed within the first 6 months after the UWL diagnosis. This appears short to me. I see many cancer patients in my clinic for whom the diagnostic workup has been longer than 6 months. I know you found the mean time between UWL and cancer diagnosis in your previous study to be 181 days (reference 2 in your paper). I nevertheless suggest you consider to use, at least in a sensitivity analysis, 12 months or longer follow-up. 

3. You used multiple imputation with 50 imputed datasets. For some variables, you had a very high proportion of missing (eg, 77% for CRP). Is 50 imputed datasets really enough? I suggest you motivate in your paper why you only used 50 imputed datasets. 

4. You use 10-fold internal cross-validation to assess model performance. To properly internally validate the model, shouldn't you validate the whole model development process including variable selection? That is, if you use 10-fold cross validation, you should in each of the 10 runs re-select the variables, fit the model and test it in the 10th fold. And then finally average the performance. You could also consider a split-sample approach. 

5. Why not test the performance of the previous models (reference 8, 9, 41, 42) in your data set? If you do that, you would have a chance to i) externally validate their models, and ii) benchmark your model against previous models. 

6. I think it is a rather major concern that your dataset/CPRD represents only 6.9% of the UK population, you have a very high proportion of missing (ie, 70-80%) for key predictors, and there are obvious concerns with over-fitting. The value and impact of your study would increase substantially with external validation. I thus strongly suggest you try to find a dataset for external validation. If this is not possible, I suggest you add a paragraph discussing these limitations and clearly state that additional research is needed before your tool can be used in clinical practice.

*** Reviewer #3: 

No comments basically 

Very well written; very important analyses, and especially the explanation box1 quite helpful for many (lay) readers

Should authors define UWL somewhere 

Typing error? Age 0+ in discussion 

Should results lead to clear GP guidelines ? (It is not advisable to have individual GPS make their own cut-offs

***

[LINK]

---

## [Decision Letter · Decision Letter 2]

14 Jun 2021

Dear Dr. Nicholson,

Thank you very much for re-submitting your manuscript "Combining simple blood tests to identify primary care patients with unexpected weight loss for cancer investigation: clinical risk score development, internal validation, and net benefit analysis." (PMEDICINE-D-21-01229R2) for review by PLOS Medicine.

I have discussed the paper with my colleagues and the academic editor and it was also seen again by two reviewers. I am pleased to say that provided the remaining editorial and production issues are dealt with we are planning to accept the paper for publication in the journal.

[LINK]

We look forward to receiving the revised manuscript by Jun 17 2021 11:59PM.   

Sincerely,

Beryne Odeny, 

Associate Editor 

PLOS Medicine

plosmedicine.org

Requests from Editors:

Thank you for your responses. Before we proceed, please address the following:

1. Given the lack of external validation, please temper the phrases “excellent discrimination” and “highly discriminative”, by removing the words “excellent” and “highly.” For example, use objective terms such as “reasonably discriminative”, “discriminative” or similar.

2. Please do not report P<0.01; report as P < 0.001.

3. Please define the abbreviations in Table 4: AST, ALT, CRP

4. Please use the "Vancouver" style for reference formatting and see our website for other reference guidelines.

5. Please include a weblink and access date for ref # 4

6. Please include line numbers in your next draft

Comments from Reviewers:

Reviewer #1: The authors have satisfactorily responded to each comment in turn, amending the manuscript accordingly.

Reviewer #2: I thank the authors for their responses to my comments and questions. The responses are scientifically satisfactory to me. That said, the major limitation of this study (ie, no external validation) remains.

[LINK]

---

## [Editor Report · Decision Letter 3]

12 Jul 2021

Dear Dr Nicholson, 

On behalf of my colleagues and the Academic Editors, Drs. Sara-Jane Dawson, Charles Swanton, and Chris Abbosch, am pleased to inform you that we have agreed to publish your manuscript "Combining simple blood tests to identify primary care patients with unexpected weight loss for cancer investigation: clinical risk score development, internal validation, and net benefit analysis." (PMEDICINE-D-21-01229R3) in PLOS Medicine.

PRESS

Sincerely, 

Beryne Odeny 

Associate Editor 

PLOS Medicine